# Effects of Metal and Metal Ion on Biomethane Productivity during Anaerobic Digestion of Dairy Manure

**Liang Yu [1,*], Do-Gyun Kim [1], Ping Ai [2], Hairong Yuan [3], Jingwei Ma [4], Quanbao Zhao [5] and Shulin Chen [1]**

[1] Department of Biological Systems Engineering, Washington State University, Pullman, WA 99164, USA
[2] College of Engineering, Huazhong Agricultural University, Wuhan 430070, China
[3] Centre for Resource and Environmental Research, Beijing University of Chemical Technology, 15 , Beisanhuan East Road, Chaoyang District, Beijing 100029, China
[4] Key Laboratory of Building Safety and Energy Efficiency, Ministry of Education, Department of Water Engineering and Science, College of Civil Engineering, Hunan University, Changsha 410082, China
[5] CAS Key Laboratory of Urban Pollutant Conversion, Institute of Urban Environment, Chinese Academy of Sciences, 1799 Jimei Road, Xiamen 361021, China
**\*** Correspondence: yuliang08@wsu.edu; Tel.: +86-509-335-7950

**Abstract:** To overcome major limiting factors of microbial processes in anaerobic digestion (AD), metal and metal ions have been extensively studied. However, there is confusion about the effects of metals and metal ions on biomethane productivity in previous research. In this study, Zn and $Zn^{2+}$ were selected as representatives of metals and metal ions, respectively, to investigate the effects on biomethane productivity. After the metals and metal ions at different concentrations were added to the batch AD experiments under the same mesophilic conditions, a Zn dose of 1 g/L and a $Zn^{2+}$ dose of 4 mg/L were found to cause the highest biomethane production, respectively. The results indicate that metal (Zn) and metal ion ($Zn^{2+}$) have different mechanisms to improve AD performance. There may be two possible explanations. To act as conductive materials in interspecies electron transfer (IET), relatively high doses of metals (e.g., 1 g/L of Zn, 10 g/L of Fe) are needed to bridge the electron transfer from syntrophic bacteria to methanogenic archaea in the AD process. As essential mineral nutrients, the AD system requires relatively low doses of metal ions (e.g., 4 mg/L of $Zn^{2+}$, 5 mg/L of $Fe^{2+}$) to supplement the component of various enzymes that catalyze anaerobic reactions and transformations. This research will provide clear insight for selecting appropriate amounts of metals or metal ions to enhance biomethane productivity for industrial AD processes.

**Keywords:** anaerobic digestion; metal; metal ion; interspecies electron transfer (IET); essential mineral nutrient; biomethane

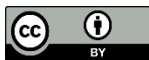

## 1. Introduction

Dairy farms have grown significantly to meet the growing demand for dairy products in the United States. The cattle population has increased steadily over the last 50 years. The total cow number reached approximately 40.6 million in 2021, and many dairy farms have been run in a concentrated animal feeding operation (CAFO). However, the environmental impacts of the CAFO have attracted increasing attention from policymakers and stakeholders, especially regarding dairy manure. It is well known that one dairy cow can produce up to 100 pounds of manure each day. The US dairy industry can generate nearly 740.95 million tons of manure a year. To improve manure management practices, the United States Environmental Protection Agency (EPA) recommends that the anaerobic digestion (AD) of dairy manure can give many environmental and economic benefits, including producing renewable energy for electricity, heating, and vehicles and reducing greenhouse gas (GHG) emissions.

AD has been widely used in many European nations. Germany is the most important country in the European biogas sector in terms of installed production and capacity, although it prioritizes the construction of small-capacity plants of 15–80 tonnes/year [1]. More than 7800 on-farm AD plants have been built using cow or pig manure as base substrate and supplemented with other agro-industrial co-substrates [2]. Compared with the European nations, the U.S. has far fewer on-farm AD plants that are working for the animal feeding operation (AFO). Although the U.S. Department of Agriculture records about 34,000 licensed dairy farms in the country, only about 200 operational dairy digesters are running to convert manure into heating, electricity, or renewable natural gas (RNG). So far, AD is not a common manure management practice in the U.S. dairy industry, mainly because of economic challenges. To overcome economic challenges, enhancing biogas yield and production rate is one of the major solutions to offset the high capital and operating costs of the digester. Microbial activity, which is an essential factor in enhancing biogas yield and production rate during AD, can be augmented by metals and metal ions.

Anaerobic microbial communities are extremely important in AD to enable diverse types and rates of reactions that eventually produce biogas. There are mainly two ways reported that can be used to increase anaerobic microbial communities' capability [3,4]. One way is to enhance electron transfer, and another way is to increase enzyme activities during anaerobic digestion. Metals and metal ions are responsible for these two ways, respectively. These two ways correspond to two different mechanisms that have been extensively investigated in AD processes. Metals [5] are used as conductive materials for direct interspecies electron transfer (DIET) between fermentation bacteria and methanogens, while metal ions [6,7] are used as essential mineral nutrients (or so-called trace elements) to supplement the component of various enzymes.

Electron transfer between microorganisms includes mediated interspecies electron transfer (MIET) and DIET. This is a key process in many reactions, especially those controlled by syntrophic relationships. MIET based on metabolite exchange between two organisms is one of the earliest discovered mechanisms of electron transfer, while DIET through extracellular electron exchange between microorganisms is another proposed mechanism that was later discovered [8]. In terms of DIET, so far, we know that two approaches to electron transfer have been discovered to form the syntrophic relationship between fermentation bacteria and methanogens [9]. The first approach is cytochromes and pili working as electron carriers in the DIET processes. The second approach is through non-biological conductive materials, such as metals, metal oxides, granular-activated carbon (GAC), biochar, carbon cloth, graphite, carbon-based nanomaterials, etc.

Among metals and metal oxides, zero-valent iron (ZVI or so-called Fe powder) and Ferrosoferric oxide are more commonly studied to enhance biomethane productivity [10]. For example, it was reported that ZVI and iron scrap of 10 g/L was the optimal dose to obtain the highest biomethane yield at 165.1 mL/g VS and 179.9 mL/g VS, respectively, after the AD of waste-activated sludge [11]. The addition of ZVI led to an increase of 14.46% in methane yield, while iron scrap caused an increase of 21.28% in methane yield. However, there was confusion about the effects of metals and metal ions in earlier research. Although many researchers claimed that different metals were studied to increase biogas production in AD processes, they actually used metal ions to conduct these experiments [12–14]. For example, Wu et al. [12] added $FeCl_2 \cdot 4H_2O$ and $ZnCl_2$ to the batch AD experiments instead of Fe and Zn powders.

Unlike metals, metal ions take on different roles in AD processes. Metal ions such as $Cu^{2+}$, $Mn^{2+}$, $Fe^{2+}$, and $Zn^{2+}$, etc., are key components of enzymes and some bacterial nucleic acids, and they are essential for the synthesis of vitamins [6]. The reported concentration of metal ions required during AD differs significantly depending on operating temperature, substrate type, digestion operating mode, and methanogenic strain. A certain amount of metal ions not only prevents the inhibition process but can also improve AD to ensure higher methane production [15]. Wang et al. reported that 5 mg/L of $Fe^{2+}$ promoted the AD of dairy wastewater to obtain the highest methane yield of 570 mL/g VS, which

was 62% higher than that of the control [7]. However, metal ions are non-biodegradable and accumulate in microbial cells. Metal ion concentrations that are higher than the toxic level can cause inhibition to anaerobic microorganisms because of the disruption of the enzyme function and structure [16]. With all that said, it is necessary to further identify the different effects between metals and metal ions in AD that can allow AD professionals and practitioners to enhance the efficiency of AD without additional equipment investment.

In this study, zinc powder and $Zn^{2+}$ ($[ZnCO_3]_2 \cdot [Zn(OH)_2]_3$) were selected as an example for metals and metal ions, respectively, to identify their different effects on the efficiency of AD with dairy manure as substrate. To the best of our knowledge, zinc powder has not been studied as a conductive material in previous research. Although many researchers claimed that they added Zn to AD processes, they used $Zn^{2+}$ compounds instead of zinc powder [17]. Zinc ion is a part of enzymes such as formate dehydrogenase (FDH), super dismutase (SODM), and hydrogenase, and it was found in remarkably high concentrations (50–630 ppm) in 10 methanogenic bacteria [6,18] (Myszograj et al., 2018; Scherer et al., 1983). We compared the process performance of AD with and without the addition of metals and metal ions in 250 mL anaerobic reactors. The modified Gompertz model was used to estimate the maximum methane production rate for the different effects of metals and metal ions.

## 2. Materials and Methods

### 2.1. Feedstock and Inoculum

Fresh dairy manure was collected from the Washington State University (WSU) Knott Dairy Center in Pullman, WA, USA, and stored at 4 °C before use. The total solids (TS) were 23.27%, while the volatile solids (VS) were 18.10%. Anaerobic sludge used as inoculum was collected from an anaerobic digester at the Pullman Wastewater Treatment Facility, WA, USA. The TS and VS of the anaerobic sludge were 1.81% and 1.31%, respectively. The anaerobic sludge was stored in a 37 °C incubator for one week before being used as inoculum to deplete the biodegradable substrate. Before being fed to digesters, dairy manure was diluted with tap water and anaerobic sludge, which resulted in a mixed liquor containing 3.12% total solids (TS) and 2.44% total volatile solids (VS).

### 2.2. Experimental Setup and Operation

A substrate/inoculum ratio of 2.0 g VS dairy manure/g VS inoculum was used for dairy manure AD experiments. All the experiments were performed in duplicate for different conditions studied. Batch experiments for the AD of dairy manure were conducted in 250 mL glass serum bottles with a liquid volume of 120 mL at 37 ± 1 °C controlled by a thermostat of the respirometry system. Before the anaerobic experiments, glass serum bottles with the mixed liquor were put in an anaerobic chamber (BACTRON™ Anaerobic/Environmental Chamber; SHEL LAB) to remove oxygen using a vacuum pump and nitrogen. The anaerobic bottle used a 0.6 mm needle to release biogas for analysis. The methane production was measured using a respirometer (Challenge Technology AER-200, Springdale, AR, USA) under standard pressure and temperature. Potassium hydroxide was used as scrubbing media to purify methane by adsorbing carbon dioxide and hydrogen sulfide so that only methane production was recorded. The methane volume production rate (mL/L/day) was calculated as the daily rate of methane produced in the liquid volume of a glass serum bottle.

### 2.3. Addition of Metal and Metal Ion

Table 1 shows the concentrations of metals and metal ions that were added in the batch experiments for the AD of dairy manure, respectively. No metals or metal ions were added to the control bottles. The initial $Zn^{2+}$ was 39.8 μg/L in the inoculum, and it was 9.99 mg/L in the raw dairy manure slurry (3.12% TS). Fe powder (<10 μm, ≥99.9% trace metals basis), zinc powder (<10 μm, ≥98% trace metals basis), and zinc hydroxide carbonate

([ZnCO$_3$]$_2$ · [Zn(OH)$_2$]$_3$, purum p.a., ≥58% Zn basis) were purchased from Sigma-Aldrich Chemicals Company, Burlington, MA, USA.

**Table 1.** Addition of metal and metal ion.

|  | Control | Fe (ZVI) | Zn | Zn | Zn | Zn | Zn |
|---|---|---|---|---|---|---|---|
| Concentration (g/L) | 0 | 6 | 0.1 | 1 | 2 | 6 | 10 |
|  |  |  | Zn$^{2+}$ | Zn$^{2+}$ | Zn$^{2+}$ | Zn$^{2+}$ | Zn$^{2+}$ |
| Concentration (g/L) |  |  | 0.004 | 0.1 | 0.5 | 1 | 2 |

*2.4. Analytical Methods*

Soluble chemical oxygen demand (sCOD), TS, VS, and ash were determined according to standard methods [19]. The pH was measured daily using a Fisher Scientific AB15 pH meter. The total ammonia nitrogen (TAN) was measured using a 2300 Kjeltec Analyzer Unit (Tecator, Perstorp Analytical, Malmö, Sweden).

*2.5. Kinetic Model Analysis*

Modified sigmoidal bacteria growth kinetic models, such as the modified Gompertz, logistic, and Richards, have been developed to evaluate the methane production process kinetics of organic wastes [20]. Among these kinetic models, the modified Gompertz model, which gave a better correlation coefficient, was used in this study [21].

$$Y = A exp\left\{-exp\left[\frac{\mu_m e}{A}(\lambda - t) + 1\right]\right\} \tag{1}$$

where $Y$ is the simulated cumulative methane yield (mL/g VS); $A$ is the simulated maximum cumulative methane yield (mL/g VS); $\mu_m$ is the maximum methane production rate (mL/g VS/day); $e$ is 2.718; $\lambda$ is the lag phase time (day); and $t$ is the digestion time (day).

*2.6. Data Statistics and Processing*

Data statistics and processing were performed using Microsoft Excel software. The Excel Visual Basic for Applications (VBA) and Solver were used for nonlinear curve fitting. The GRG Nonlinear engine was used to search for $\mu_m$ to meet the minimum mean-squared error between experimental methane yield and predicted methane yield. Significant differences in methane yield among the control, Fe, and different zinc and zinc ion concentrations were determined by performing an analysis of variance (ANOVA) at $p < 0.05$ using the Tukey test. Minitab 17 software was used in this study.

**3. Results and Discussion**

*3.1. Effect of Metal and Metal Ion on Methane Yield and Volume Production Rate*

Methane yield and volume production rate are key performance metrics for AD that should be met not only at a laboratory (lab)-scale but also at subsequent pilot- and commercial-scale biogas plants [22]. Methane yield directly affects the production cost because it determines the amount of dairy manure needed for methane production. The methane volume production rate determines the overall anaerobic digester volume needed for a given output, affecting the direct fixed capital costs and associated annual operation costs through annual depreciation [23]. Figures 1 and 2 show the effect of metals and metal ions on methane yield and volume production rate. To investigate the varying effects of metals and metal ions on the AD process, we studied zinc and zinc ions as a representative example. Fe was used for comparison. The control was conducted for mesophilic AD without the addition of metal or metal ion. Zinc ion is involved in many aspects of cellular metabolism. More than 200 enzymes require zinc ion as a functional component, and these enzymes affect most major metabolic processes. It plays an essential role in enhancing immune function, protein and DNA synthesis, wound healing, and cell signaling and division [24]. However, zinc ion

is often mistaken for zinc, which is used as a conductive material to promote electron transfer between anaerobic bacteria and archaea [5,25,26].

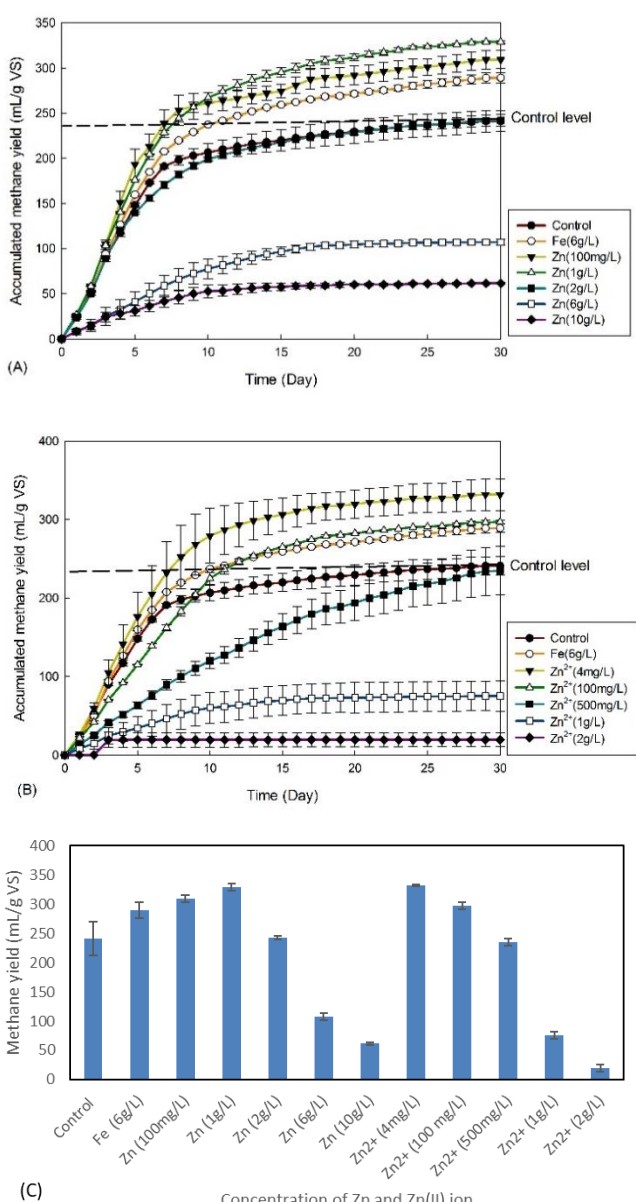

**Figure 1.** Effect of zinc and zinc ion on (**A**) accumulated methane yield and (**B**) methane yield (**C**) average methane production rate (mL/L/day).

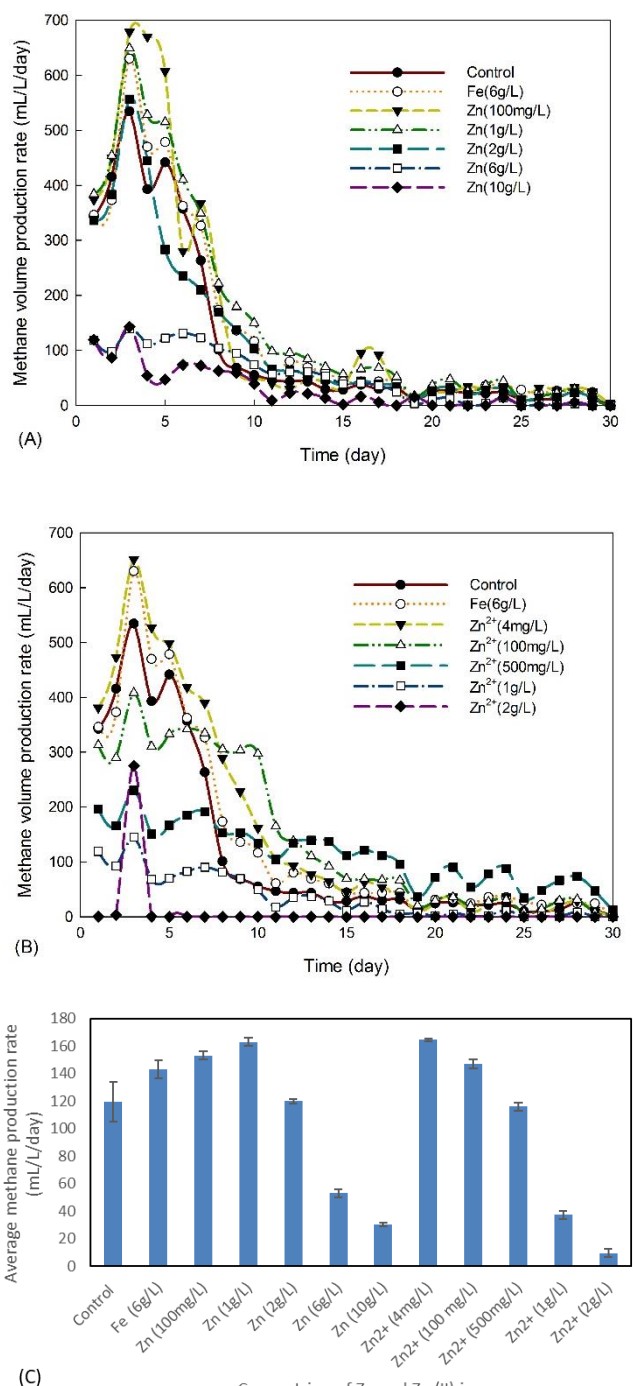

**Figure 2.** Effect of zinc and zinc ion on (**A**) methane volume production rate and (**B**) average methane volume production rate (**C**) average methane production rate (mL/L/day).

Figure 1 gives a comparison between zinc and zinc ion for their effects on methane yield. The experimental results of the change in accumulated methane yield with time for each batch experiment with and without the addition of metal and metal ion are shown in Figure 1A. In this figure, different trends caused by different doses of Zn and $Zn^{2+}$ can be observed. When the Zn dose was at less than 6 g/L, the accumulated methane yields went up rapidly in the first 6 days. After day 6, the increase of the accumulated methane yields slowed down and then gradually flattened out. Similar trends of the change in accumulated methane yield with time occurred in the control, the comparison of Fe dose at 6

g/L, and $Zn^{2+}$ with a dose of less than 100 mg/L. However, when the $Zn^{2+}$ dose was at 100 mg/L or the Zn dose was at 6 g/L, the steep rise in the accumulated methane yields tended to be a gentle increase. After the $Zn^{2+}$ dose was more than 100 mg/L or the Zn dose was more than 6 g/L, the accumulated methane yields increased more and more slowly, suggesting that a high amount of metal or metal ion caused inhibitory, stimulatory, and even toxic effects on anaerobic digestion [27,28]. When the $Zn^{2+}$ dose reached 2 g/L, a long lag phase for biomethane production occurred, and the maximum methane yield was only 19.42 mL/g VS, which is 1/11 of that of the control.

The methane yields with and without the addition of metals and metal ions are shown in Figure 1B. The methane yield from the control experiment was 241.18 mL/g VS, while the methane yield from the batch experiment with 6 g/L of Fe was 289.06 mL/g VS, which was 19.85% higher than that of the control. The batch experiment with 4 mg/L of $Zn^{2+}$ achieved the highest methane yield at 331.85 mL/g VS, which was 37.59% higher than that of the control. The result for the optimum amount of $Zn^{2+}$ is similar to that of the previous research at 2.1 mg/L (Wu et al., 2016). A possible reason is that zinc ion is an essential mineral nutrient for the component of enzymes and it accumulates in microbial cells (Zandvoort et al., 2003). A very small amount of zinc ion is required for microbial cells. This is why it is called a trace element. High metal ion concentration can disrupt the enzyme function and structure. Therefore, $Zn^{2+}$ with a dose of 1 g/L caused a significant drop in methane yield, which was 75.26 mL/g VS and 68.8% less than that of the control. By contrast, the batch experiment with 1 g/L of Zn achieved the highest methane yield at 328.88 mL/g VS, which was 36.36% higher than that of the control. A possible reason is that zinc works as a conductive material for electron transfer. Conductive materials exist outside of the microbial cells so that they are less harmful to microbes. A relatively high dose of Zn is needed to facilitate electron transfer. However, metal ions can be released from metals because of electrochemical corrosion and oxidation, dissolution of surface oxides, etc. (Hedberg and Odnevall Wallinder 2016). Higher doses of metals may cause higher concentrations of released metal ions. This phenomenon can be used to explain why the methane yield decreased after the dose of Zn was more than 1 g/L. Therefore, our experimental findings indicate that metals and metal ions exert varying effects on AD, all of which contribute to improving biogas production.

Figure 2 gives a comparison between zinc and zinc ion for their effects on methane volume production rate. The experimental results of the change in methane volume production rate with time for each batch experiment with and without the addition of metals and metal ions are shown in Figure 2A. A sharp rise in the methane volume production rate was observed under each condition, except the $Zn^{2+}$ dose of 2 g/L, which caused a long lag phase. The highest methane volume production rates of all batch experiments were achieved on the third day. After day 3, a steep fall in the methane volume production rates was observed when the Zn dose was less than 6 g/L, and then the line flattened out gradually. Similar trends of the change in methane volume production rate with time occurred in the control, the comparison of Fe dose at 6 g/L, and $Zn^{2+}$ with a dose of less than 100 mg/L. This indicates that the anaerobic microorganisms were more effective and efficient in using up the substrates under the conditions. However, when the $Zn^{2+}$ dose was more than 100 mg/L or the Zn dose was more than 6 g/L, the steep fall in the methane volume production rates tended to be a gentle decrease. The $Zn^{2+}$ dose of 2 g/L caused the methane volume production rate to drop to zero, suggesting that methanogens were completely inhibited.

The average methane volume production rates with and without the addition of metals and metal ions are shown in Figure 2B. The average methane volume production rate from the control experiment was 119.51 mL/L/day, while the average methane volume production rate from the batch experiment with 6 g/L of Fe was 143.25 mL/L/day, which was 19.86% higher than that of the control. The batch experiment with 1 g/L of Zn achieved the highest average methane volume production rate at 163.05 mL/L/day, which was 36.43% higher than that of the control because of the zinc. By contrast, the batch experiment with 4 mg/L of $Zn^{2+}$ achieved the highest average methane volume production

rate at 164.51 mL/L/day, which was 37.59% higher than that of the control because of the zinc ion. Nevertheless, $Zn^{2+}$ with a dose of 1 g/L caused a significant fall in the average methane volume production rate, which was 37.31 mL/L/day and 68.78% less than that of the control. Therefore, the addition of metals and metal ions into anaerobic digesters can affect both methane yield and volume production rate.

### 3.2. Effect of Metal and Metal Ion on Soluble COD and VS

sCOD and VS represent organic substrates that microorganisms can uptake [29]. These two parameters are often used to estimate the biodegradability of organic matter because they are easily measured in practice. sCOD is commonly used to evaluate hydrolysis, which is an essential step in AD [30]. During hydrolysis, the most complex particulate organic matter (polymers) is transformed into dissolved simple materials by hydrolytic bacteria, such as *Clostridia, Micrococci, Bacteroides, Butyrivibrio, Fusobacterium, Selenomonas, Streptococcus*, etc. [31]. They secret different hydrolyzing enzymes, such as cellulase, cellobiase, xylanase, amylase, protease, lipase, etc., to accelerate hydrolytic reactions.

Figure 3A shows the changes in sCOD concentration with the different doses of zinc and zinc ion in the batch AD experiments under mesophilic conditions. The decrease in sCOD from the control experiment was 55.63%, while the decrease in sCOD from the batch experiment with Fe of 6 g/L was 62.09%, compared with the initial sCOD concentration. This shows that there was a negative correlation between biomethane production and sCOD concentration. However, the Zn doses of 100 mg/L, 1 g/L, and 2 g/L did not cause the sCOD concentration to change significantly ($p > 0.05$) during the batch AD experiments. The corresponding decreases in sCOD concentrations were 62.58%, 60.76%, and 62.58%, respectively. This is because, besides the initial sCOD concentration, sCOD could still be generated by solid organic matter during hydrolytic reactions in the batch AD experiments. Therefore, sCOD removal could not be calculated based on the initial and final sCOD concentrations. There were even increases in sCOD concentrations when the Zn dose was at 6 g/L or 10 g/L. The corresponding sCOD concentrations were 9.77% and 62.75%, respectively, higher than the initial sCOD concentration. A similar situation was observed in the addition of zinc ion during the batch AD experiments. The sCOD removals from the batch experiment with 4 mg/L of and 100 mg/L of $Zn^{2+}$ were 59.93% and 57.45%, respectively. By contrast, the increases in sCOD concentration from the batch experiment with 500 mg/L, 1 g/L, and 2 g/L were 11.26%, 68.87%, and 53.48% of $Zn^{2+}$, respectively, compared with the initial sCOD concentration. This indicates that hydrolytic bacteria are more resistant to the inhibition caused by an overdose of metals and metal ions than methanogens [27].

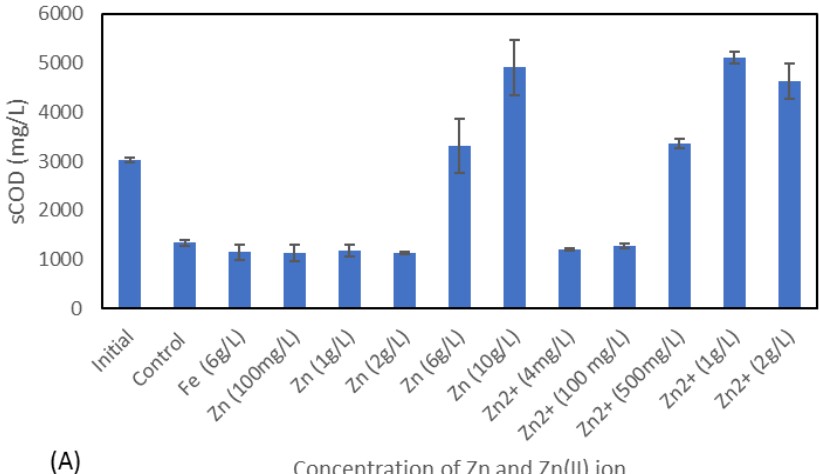

(A)

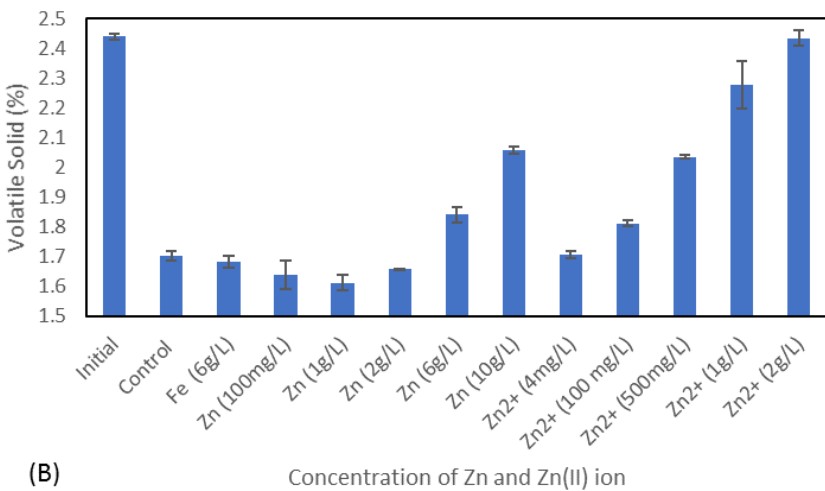

(B)

Concentration of Zn and Zn(II) ion

**Figure 3.** Effect of zinc and zinc ion on (**A**) soluble COD and (**B**) VS.

The VS in dairy manure is not highly degradable compared to that of other animal manures, owing to the high digestion efficiency of the rumen system in cows, along with their fibrous diet [32]. Hill reported the VS reduction for dairy manure at only 23% compared to 63% for both swine manure and poultry manure [33]. In this study, as shown in Figure 3B, the VS reduction for the control experiment was 30.24%, while the VS reduction for the batch experiment with 6 g/L of Fe was 31.10%, compared with the initial sCOD concentration. The VS reduction between the control and the batch experiment with 6 g/L of Fe did not exhibit a proportional increase with the increase in methane production because the methane yield from the Fe dose of 6 g/L was 19.85% higher than that of the control. The addition of zinc and zinc ion also caused the same trend of VS reduction in the batch AD experiments. The batch experiment with 1 g/L of Zn achieved the highest VS reduction at 33.97%. By contrast, the batch experiment with 4 mg/L of $Zn^{2+}$ achieved the highest VS reduction at 30.14%. Nevertheless, $Zn^{2+}$ with a dose of 1 g/L caused a significant drop in VS reduction, which was 6.66%. Therefore, an anaerobic microbial community obviously requires different amounts of zinc and zinc ion that allow it to achieve the highest activity and produce more biogas.

### 3.3. Effect of Metal and Metal Ion on TAN and pH

TAN and pH are essential process parameters that can be monitored and controlled to maintain an anaerobic digester's stable operation. TAN is the total amount of nitrogen in the forms of free/unionized ammonia ($NH_3$) and ionized ammonium ($NH_4^+$) in wastewater. Ammonia is the end-product that is produced from the anaerobically digested proteins, urea, and nucleic acids [34]. Figure 4A shows the changes in TAN concentration with the different doses of zinc and zinc ion in the batch AD experiments under mesophilic conditions. In contrast with the initial TAN concentration, almost similar increases in the final TAN concentrations caused by the batch AD processes with and without the addition of metal and metal ion were observed in the figure. This indicates that similar amounts of ammonium and ammonia from dairy manure were released during hydrolysis and acidogenesis according to AD reaction pathways [35]. Previous research reported that inhibition occurred, causing a significant reduction in methane production, if TAN was more than 1700 mg/L [36]. As shown in Figure 4A, the final TAN concentrations were less than 890 mg/L, suggesting that ammonia did not cause inhibition in the batch AD experiments. It was high doses of zinc and zinc ion that caused inhibition and reduced methane production. The experimental results further confirm that hydrolytic bacteria and acidogens are believed to be more resistant to heavy metal toxicity than methanogens [27] (Chen et al., 2014).

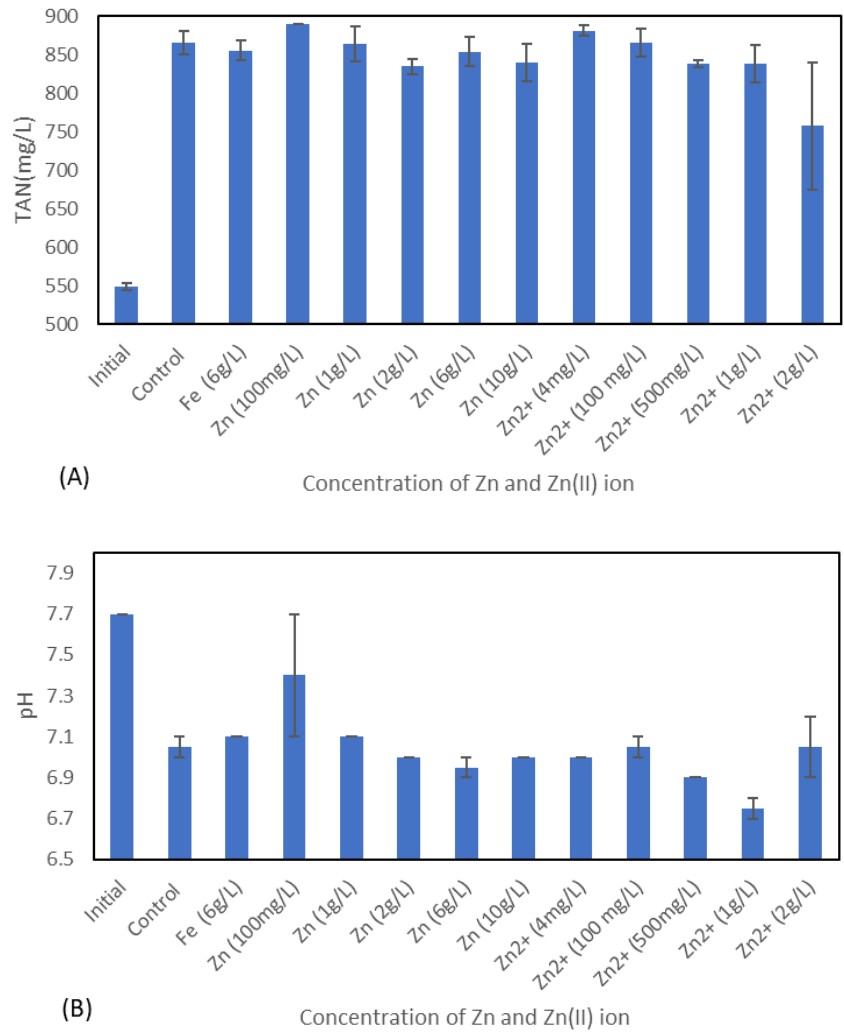

**Figure 4.** Effect of zinc and zinc ion on (**A**) TAN and (**B**) pH.

pH is an expression of the intensity of the alkaline or acidic condition of aquatic solutions. Commonly, anaerobic digesters can operate stably at pH levels from 6.3 to 7.8 without suffering any reduction in methane volume production rate. Methanogens in AD systems tend to be most active in a pH range between 6.7 and 7.4 [36]. Figure 4B shows the changes in pH with the different doses of zinc and zinc ion in the batch AD experiments under mesophilic conditions. In contrast with the initial pH, almost similar decreases in the final pH caused by the batch AD processes with and without the addition of metal and metal ion were observed in the figure. When dairy manure is anaerobically digested in digesters, organic particles are degraded to volatile fatty acids (VFAs). These organic acids accumulate before the methanogens can use them. Therefore, the pH initially tends to decrease. However, dairy manure usually contains enough alkalinity to buffer these organic acids and prevent them from decreasing pH too much [36]. Furthermore, the final pH levels caused by the batch AD processes with and without the addition of metal and metal ion were between 6.75 and 7.4, suggesting that the pH environment was favorable for methanogenesis. The methane yield and methane volume production rate were reduced in high doses of zinc and zinc ion because methanogens are sensitive to toxic substances, and those methanogens using acetate were more susceptible to toxic substances than methanogens using $H_2/CO_2$ [37].

### 3.4. Kinetic Analysis of Methane Production

The Gompertz model is one of the theoretical kinetic models for the prediction of methane production from anaerobic digestion and the exploration of the interaction mechanism [20]. The modified Gompertz equation has been widely used to calculate the kinetic behavior for conductive materials' electron transfer and the effects of trace elements' addition [5,12]. Previous research reported that Gompertz and logistic models have the best coefficients of determination compared to other functions, such as Transference, First Order, and Richard [20]. The input data (maximum cumulative methane yield, maximum methane production rate, and lag phase time) for these models are accessible. In this study, a modified Gompertz model was used to fit the batch AD experimental results with and without the addition of metals and metal ions. It has been frequently used to describe microbial growth.

Figure 5 shows the modified Gompertz model fitting results. As shown in Figure 5A, the correlation coefficient ($R^2$) was used to evaluate the goodness of the modified Gompertz model fitting. Except for the $Zn^{2+}$ dose of 2 g/L, high $R^2$ ranging from 0.967 to 0.999 was obtained under the other conditions, indicating the prediction from the modified Gompertz equation fit well with the experimental data. Even for the $Zn^{2+}$ dose of 2 g/L, an $R^2$ of 0.727 was obtained. The reason might be that a lag phase (2 days) and severe inhibitions occurred under this condition, making the biochemical reactions more complicated. Therefore, the microbial growth curve was not in good shape. Nevertheless, according to the measure of the goodness of fit of the model, an $R^2$ above 0.7 would generally be seen as showing a good correlation between the variables. The maximum methane production rates ($\mu_m$) under the corresponding conditions were estimated using the GRG Nonlinear engine in the Excel Solver. $\mu_m$ is calculated by dividing the VS added into the digester instead of the digester's effective volume. The same trend for $\mu_m$ as the methane yield in Figure 1B and the average methane volume production rate in Figure 2B can be observed in Figure 5A. The highest $\mu_m$ for a Zn dose of 1 g/L was 32.078 mL/g VS/day, while the highest $\mu_m$ for a $Zn^{2+}$ dose of 1 g/L was 33.822 mL/g VS/day. Excellent correlations ($R^2 > 0.95$) between the experimentally measured and model-predicted values for accumulated methane yields under the control, Zn dose of 1 g/L, and $Zn^{2+}$ dose of 4 mg/L were observed in Figure 5B–D, respectively. Since the modified Gompertz model is regarded as one of the best methods that allow the prediction of the methane production and characterization of methanogen potentials of substrates [20], it can be used to further indicate that metals and metal ions exert different effects on AD to improve biogas production.

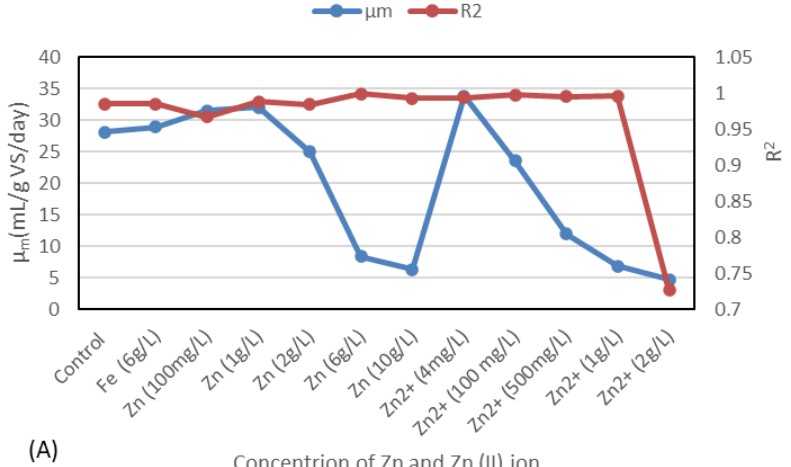

(A)

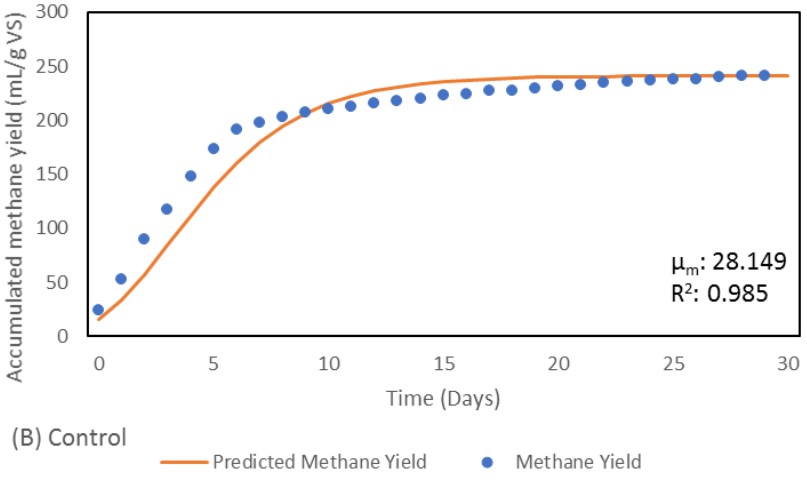

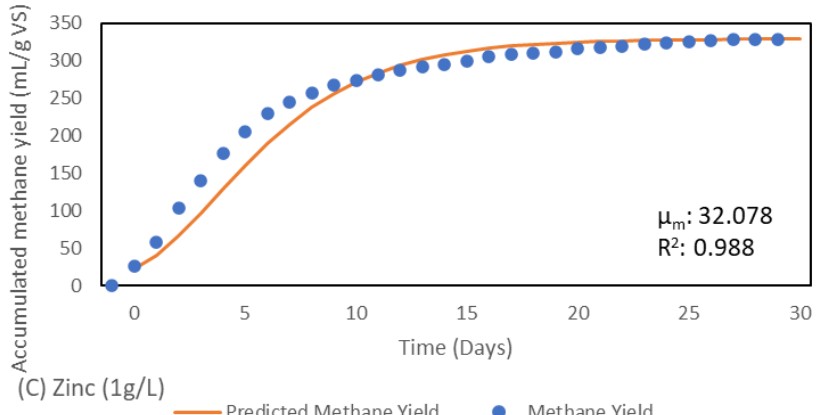

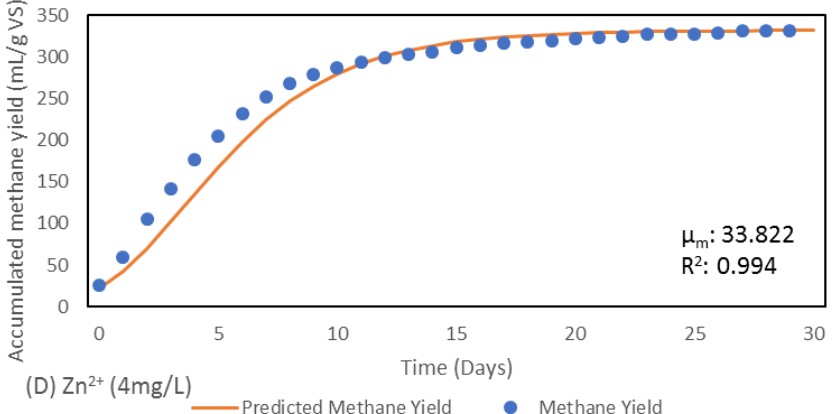

**Figure 5.** Modified Gompertz model fitting results for accumulated methane yield (**A**) μm and $R^2$; (**B**) control; (**C**) Zn (1 g/L); and (**D**) $Zn^{2+}$ (4 mg/L).

## 4. Conclusions

In this study, batch experiments were used to investigate the effects of metals and metal ions on the AD process of dairy manure to enhance biomethane production. A zinc powder amount of 1 g/L was found to be the optimum concentration, which caused the highest methane yield of 328.88 mL/g VS, which was 36.36% higher than that of the control. The methane yield increased with the increase in the Zn dose ranging from 100 mg/L to 1 g/L, while

it decreased with the increase in the Zn dose ranging from 1 g/L to 10 g/L. By contrast, 4 mg/L of $Zn^{2+}$ was found to be the optimum concentration, which caused the highest methane yield of 331.85 mL/g VS, which was 37.59% higher than that of the control. Methane yield decreased with the increase in the $Zn^{2+}$ dose ranging from 4 mg/L to 2 g/L. The modified Gompertz model provides a good model fit for the measured accumulated methane yields of dairy manure during AD with and without the addition of metals and metal ions. The estimated maximum methane production rates ($\mu_m$) using the modified Gompertz equation can serve as an indicator that metals and metal ions operate through distinct mechanisms to enhance biogas production. Further research is needed to elucidate the specific mechanisms underlying the enhancement of methane production by metal-related electron transfer and metal ion–related essential mineral nutrients that can be used for industrial AD facilities.

**Author Contributions:** Conceptualization, L.Y. and D.-G.K.; methodology, L.Y., D.-G.K., P.A., H.Y., J.M. and Q.Z.; software, L.Y.; validation, D.-G.K. and L.Y.,; formal analysis, P.A., H.Y., J.M., Q.Z.; investigation, D.-G.K.; resources, L.Y., P.A., H.Y., J.M., Q.Z. and S.C.; data curation, D.-G.K.; writing—original draft preparation, S.C.; writing—review and editing, P.A., H.Y., J.M., and Q.Z.; visualization, L.Y.; supervision, L.Y.; project administration, S.C.; funding acquisition, L.Y. and S.C. All authors have read and agreed to the published version of the manuscript.

**Funding:** This material was based on work supported by the U.S. Department of Energy's Office of Energy Efficiency and Renewable Energy (EERE) under the Bioenergy Technologies Office Award Number DE-EE0008808/0001. The views expressed herein do not necessarily represent the views of the U.S. Department of Energy or the United States Government. This work was also partially supported by the Agriculture and Food Research Initiative Competitive Grants Program [grant no. 2022-67019-36486/project accession no. 1028107] from the USDA National Institute of Food and Agriculture. This activity was funded, in part, with an Emerging Research Issues Internal Competitive Grant from the CAHNRS Office of Research at Washington State University, College of Agricultural, Human, and Natural Resource Sciences.

**Institutional Review Board Statement:** Not applicable.

**Informed Consent Statement:** Not applicable.

**Data Availability Statement:** All data were included in the manuscript.

**Conflicts of Interest:** The authors declare no conflict of interest.

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
