# Peer review of "Effects of Metal and Metal Ion on Biomethane Productivity during Anaerobic Digestion of Dairy Manure"

_fermentation, doi:10.3390/fermentation9030262_

Round 1

Reviewer 1 Report

Dear author,

Thank you for your work. However, the article should be revised and some additional information must be added.

Topic: I think the real topic would be the effect of Zn and not only metals

Abstract: I miss some results (numbers) in LL. 24-29 in the abstract.

LL. 23 Really? I missed this point. Different results yes, but different mechanism?

LL. 45 which size is „small-capacity plants“?

LL. 99-106: In the introduction i miss scientific sources about the effect of Zn. You only describe the effects of Fe and suddenly you change to Zn. Is there no previous work?

2.1 Do you have information about the trace elements in the inoculum and the substrates? I think it would be help to understand the results

LL. 115 Where is the anaerobic sludge from?

2.2 Did you calculate the values to standard pressure and temperature? Did you measure the methane yield of the incoulum yourself? How many repetitions did you make?

LL. 149: VS is not mention in the formula

 LL. 162-169: the production rate should be explained in material and methods

LL. 169: Figure 1 and 2 is some page behind your annoucement. You should change the structure

LL. 173-179: Sounds like introduction

If you only talk about Zn please don’t change everytime to „metal“. It is confusing.

LL. 195: How can I see it? I am confused. Again, without the trace element concentration in the original substrate, the results are difficult to interpret

LL: 197-207: Can you explain why the results look loke that?

Figure 1: Zn 2 g/l and control look quite similar (both light blue).

Please add the standard deviation at every accumulated Methane yield

Figure 2: How you calculate the methane production rate? By gompertz?

LL. 249-252: In an anaerobic digestion process, sCOD is degraded.  It does not „seem...", it is so!

LL. 268-270: Please check the acid concentration. This should underline your thesis or not. Is it possible that the standard deviation is higher in this case instead of the methane yield? It could be also a reason fort hat behaviour.

Figure 5: a lag phase of zn2+ should be good fitted by Zn2+. I’m wondering about your results.

LL. 355: How you like to use the gompertz  model in future for that?

Conclusion: You need the starting concentration to get an idea how much Zn should be added in the process!

Reviewer 2 Report

The manuscript presents a study on the impact of zinc particles and zinc ions on the biogas generation during anaerobic digestion processes. Different concentrations are tested and the comparison between zinc particles and zinc ions is discussed.

The manuscript addresses an interesting question which can be of interest for the scientific community.

Nevertheless, in my opinion the data do not support enough the claim that these two forms of zinc play a different role in the AD process. Despite the fact that this concept is stated multiple times throughout the manuscript, the data seems to imply the opposite: zinc particles and zinc ions show a similar behavior (i.e. enhancement of methanogenesis until a certain threshold and then inhibition) but at different concentrations. Additional discussion would be necessary to support this hypothesis. Unfortunately, the modeling part does not add much to the discussion and does not prove or disprove what the authors are claiming.

In the introduction section, the authors should discuss the addition of metal particles in AD reactors if there are reports of those, or clearly state that it was never attempted.

Furthermore, the clarity of the manuscript can be greatly improved, especially on a linguistic level. Few examples follow:

·       Lines 71 – 72: the word “mechanism” is not pertinent

·       Line 83: what “indeed” stand for?

·       Line 94-95 are very unclear. Please rephrase

·       Line 124: a vacuum “pump” maybe?

·       Line 163: metrics cannot be “satisfied”

·       Line 249 and 296: “affected” is not a good word referring to concentrations

·       Line 253: it is more of an inverse correlation!

·       Line 319: remove “up”

·       Line 332: remove “the”

Finally, the figures can be improved to facilitate the readability:

·       Fig 2.A is quite difficult to read. Maybe use different colors and/or separate the data in two different graphs?

·       Different colors should be used to visualize the different treatments (i.e. Control, Iron, zinc particles and zinc ions)

·       The control level could be indicated with a horizontal line, or alternatively the data could be represented as percentage of increase/decrease with respect to the control

For these reasons, I recommend the publication of this manuscript only after major revisions.

Round 2

Reviewer 1 Report

Dear Author,

Comment 10: LL. 162-169: the production rate should be explained in material and methods.
Response: It has been explained in material and methods 2.2 as “The methane volume production
rate (mL/L/day) was calculated as the daily rate of methane produced in the liquid volume of a
glass serum bottle.” “Methane production rate” was changed to “methane volume production rate”
to distinguish it from “maximum methane production rate (mL/g-VS/day)”

is it the average value of all days? Your retention time will strongly effect this value. Is it a usefull value?

Comment 16: Please add the standard deviation at every accumulated Methane yield.
Response: The standard deviation has been added.

You didn't do it yet. Please change it.

Comment 22

Please add the answer in the conculusion text

Please check your units (e.g. /VS-1)

Reviewer 2 Report

The authors addressed most of the points raised during the last revision. However, my major comment was not addressed properly. The data do not clarify or confirm that metals and metal ions take part in different mechanisms like it is stated in the abstract (line 24), in the introduction (line 113) and in the discussion (line 193 and 247). It is more of an indication.

The authors point out that previous literature already discussed this topic. Nevertheless, the presented data do not provide much additional evidence to support the statement. It is unclear if the metal particles are partially dissolved providing Zn ions to the medium, if they participate in the DIET, how the microbial communities are affected by these two amendments, if they provide synergistic, antagonistic or additive effects, etc.

The modeling part still does not add much to the overall discussion.

For these reasons, major revisions are still required in order to publish this manuscript.
